# A linear model to derive melt pond depth on Arctic sea ice from hyperspectral data

Marcel König[1], Natascha Oppelt[1]

[1]Department of Geography, Kiel University, Kiel, 24118, Germany

*Correspondence to*: Marcel König (koenig@geographie.uni-kiel.de)

**Abstract.** Melt ponds are key elements in the energy balance of Arctic sea ice. Observing their temporal evolution is crucial for understanding melt processes and predicting sea ice evolution. Remote sensing is the only technique that enables large-scale observations of Arctic sea ice. However, monitoring melt pond deepening in this way is challenging because most of the optical signal reflected by a pond is defined by the scattering characteristics of the underlying ice. Without knowing the

influence of melt water on the reflected signal, the water depth cannot be determined. To solve the problem, we simulated the way melt water changes the reflected spectra of bare ice. We developed a model based on the slope of the log-scaled remote sensing reflectance at 710 nm as a function of depth that is widely independent from the bottom albedo and accounts for the influence of varying solar zenith angles. We validated the model using 49 in situ melt pond spectra and corresponding depths from shallow ponds on dark and bright ice. Retrieved pond depths are accurate ($RMSE = 2.81$ cm; $nRMSE = 16\,\%$) and highly

correlated with in situ measurements ($r = 0.89$; $p = 4.34$e-17). The model further explains a large portion of the variation in pond depth ($R^2 = 0.74$). Our results indicate that our model enables the accurate retrieval of pond depth on Arctic sea ice from optical data under clear sky conditions without having to consider pond bottom albedo. This technique is potentially transferrable to hyperspectral remote sensors on UAVs, aircraft and satellites.

## 1 Introduction

Melt ponds on sea ice are key elements for the Arctic energy budget. They are a main driver of the ice-albedo-feedback mechanism (Curry et al., 1995) and affect the mass and heat balance of sea ice (e.g. Flocco et al., 2012; Perovich et al., 2009). Observations of pond evolution can be linked to observations of sea ice, ocean and atmosphere (e.g. Inoue et al., 2008; Polashenski et al., 2012; Webster et al., 2015), for validation of ice and climate models (e.g. Flocco et al., 2012) and future sea ice prediction (e.g. Schröder et al., 2014). In the context of climate change it is therefore important to increase our

understanding of how melt ponds on sea ice change (Lee et al., 2012).

Recent efforts were made to observe the evolution of melt pond fraction with satellite data (e.g. Istomina et al., 2015a, 2015b; Rösel et al., 2012; Tschudi et al., 2008; Zege et al., 2015) but few studies investigated melt pond depth despite its relevance for many applications. Melt pond depth is a parameter in the Los Alamos sea ice model CICE (Flocco et al., 2012; Hunke et al., 2013) and the ECHAM5 general circulation model (Pedersen et al., 2009). Lecomte et al. (2011) used pond depth to

parameterize melt pond albedo in a snow scheme for the thermodynamic component of the Louvain-la-Neuve sea ice model. Holland et al. (2012) related pond water volume to surface meltwater fluxes in the Community Climate System Model, version 4; and Palmer et al. (2014) used melt pond depths to model primary production below sea ice. Liu et al. (2015) point out that climate models and forecast systems that account for realistic melt pond evolution "seem to be a worthy area of expanded research and development" (Liu et al., 2015) and question the suitability of statistical forecasting methods in the context of the

changing Arctic pointing towards the need for regular observations with large spatial coverage.

Synoptic observations of melt pond evolution are only possible with satellite remote sensing. Optical sensors with an adequate spatial resolution that operate in the visible (VIS) and near infrared (NIR) wavelength regions enable monitoring of pond water characteristics. The reflected optical signal from melt ponds without ice cover contains information on the pond water, the pond bottom, i.e. underlying ice and skylight reflected at the water surface.


Some studies investigated the potential to map the bathymetry of melt ponds with optical data in supraglacial lakes on the Greenland ice sheet. Tedesco and Steinar (2011) used the model of Philpot (1989) for optically shallow water and resampled hyperspectral reflectance measurements from below the water surface to Landsat and MODIS bands in order to explore its capability to derive the depth of a supraglacial lake. Due to the strong absorption of water in the near infrared, they limited the

data range to 450 – 650 nm and excluded depth measurements < 1 m "because of the relatively small sensitivity of the reflectance data in the Landsat and MODIS blue and green bands to shallow waters" (Tedesco and Steiner, 2011). In comparison with shallow water sonar measurements, they underestimated depth by -23.7% and -42.7% for Landsat bands 1 and 2, respectively. Legleiter et al. (2014) used hyperspectral remote sensing reflectance measurements above the water surface to map the bathymetry of supraglacial lakes and streams. They used an optimal band ratio analysis to find suitable band

combinations for calibrating an empirical model based on field measurements on the Greenland ice sheet. A model based on two bands in the yellow-orange wavelength region resulted in a $R^2$ of 0.92 and a standard error of 0.47 m for depths ranging between 0.31 m and 10.45 m. While this accuracy may be sufficient for glacial lakes, the maximum depth of ponds on sea ice is restricted by its thickness and therefore seldom exceeds 1 m (e.g. Morassutti and Ledrew, 1996; Perovich et al., 2009).

The color of melt ponds on sea ice ranges from bright blue to almost black and is primarily defined by the scattering and, to a

lesser degree, by the absorption characteristics of the pond bottom (Lu et al., 2016, 2017). Different radiative transfer models for melt ponds on sea ice exist but their capability to derive pond depth varies. Lu et al. (2016, 2017) developed a two-stream radiative transfer model to retrieve pond depth and the thickness of the underlying ice from RGB images but did not find a clear relationship between simulated and measured pond depth using the data by Istomina et al. (2016). To our knowledge, the most accurate model is the one presented in Malinka et al. (2018) resulting in a $R^2$ of 0.62 ($N = 26$) for in situ pond depths

between 6 cm and 50 cm acquired under different illumination conditions. Their analytical two-stream radiative transfer model links the spectral albedo of ponds between 350 and 1300 nm at various sky conditions to pond depth and transport scattering coefficient and thickness of the bottom ice. Fitting these parameters during inverse computation of in situ datasets from three

field campaigns accurately reproduced in situ albedo spectra (relative root mean square difference ($rRMSD$) < 1.5%) but pond depth retrieval was more uncertain ($rRMSD = 65\%$).


We hypothesize that instead of using the entire spectrum, selecting bands in the near infrared wavelength region improves the retrieval of pond depth on sea ice from optical data. The penetration depth of light into water is highest in the blue region of the electromagnetic spectrum and decreases with increasing wavelength, i.e. with increasing wavelength the influence of the water column's attenuation on the optical signal increases (Pope and Fry, 1997). Mapping the bathymetry of supraglacial lakes

with a two-band model is challenging, because the attenuation of water is wavelength dependent and the range of depths is wide. For shallow ponds on sea ice Morassutti and Ledrew (1996) stated that the influence of water absorption on the pond albedo increases towards the NIR wavelength region. Lu et al. (2016) found that pond albedo significantly depends on pond depth in the wavelength region between 600 nm and 900 nm. In this paper, we therefore present a linear pond depth model for Arctic sea ice, based on the absorption of near infrared light in water from hyperspectral optical measurements under clear sky

conditions.

## 2 Methods

We use spectral data of bare ice surfaces to simulate melt pond spectra for model development, and validate the model with in situ melt pond measurements acquired during RV *Polarstern* cruise PS106 in summer 2017.

### 2.1 Observational data

We used two instrument setups for acquisition of optical data. For most measurements, we used a combination of two Ocean Optics STS-VIS spectrometers (Ocean Optics Inc., USA). One spectrometer pointing downwards and equipped with a 1° fore optic; the other pointing upwards and equipped with a cosine collector. Both instruments cover the wavelength region from ~ 340 nm to ~ 820 nm with a spectral resolution of 3.0 nm (Ocean Optics, 2019). We used a Labsphere Spectralon 99 % diffuse reflectance standard (Labsphere Inc., USA) as white reference and applied the data from the second spectrometer to correct

the reflectance spectra for changes in downwelling irradiance. For each measurement, we computed the average of 30 single spectra. Both instruments were mounted to the end of a 1 m long pole to avoid influences of the polar clothes on the measurements. We also attached a camera to the setup to take photographs of each measurement site (Figure 1).

Some of the data used in this study were acquired within the scope of an angle-resolving BRDF experiment. For these measurements, we used an Ibsen Freedom VIS FSV-305 spectrometer (Ibsen Photonics A/S, Denmark) with a spectral

resolution of 1.8 nm covering the wavelength range from ~ 360 nm to ~ 830 nm (Ibsen Photonics, 2019). The spectrometer was equipped with an optical fiber and a 1° fore optic that were attached to a field goniometer (Figure 2). We used the above-mentioned Spectralon panel as white reference after each azimuthal scan and computed an average reflectance from 20 spectra.

The quantity measured with both spectrometer setups is the remote sensing reflectance $[sr^{-1}]$ ($R_{rs}$) above the water surface:

$$R_{rs} = \frac{L_u}{E_d},$$
(1)

where $L_u$ is upwelling radiance $[W/(m^2\ nm\ sr)]$ measured by the downwards-pointing sensor and $E_d$ is downwelling irradiance $[W/(m^2\ nm)]$ derived from the Spectralon measurement as

$$E_d = \frac{L_S \cdot \pi}{R_S},$$
(2)

where $R_S$ is the isotropic reflectance of the Spectralon panel, and $L_S$ is a radiance measurement $[W/(m^2\ nm\ sr)]$ of the Spectralon panel.

### 2.1.1 Ice spectra

On 15 June 2017, we used the Ocean Optics setup to collect spectra from three bright and one dark bare ice surface (Gege et al., 2019) that were missing the typical surface scattering layer (Figure 1A, B). We therefore assume that their optical properties are comparable to pond bottoms. Illumination was diffuse and stable indicated by the negligible standard deviation of the 30 spectra contained in one measurement (Figure 1C).

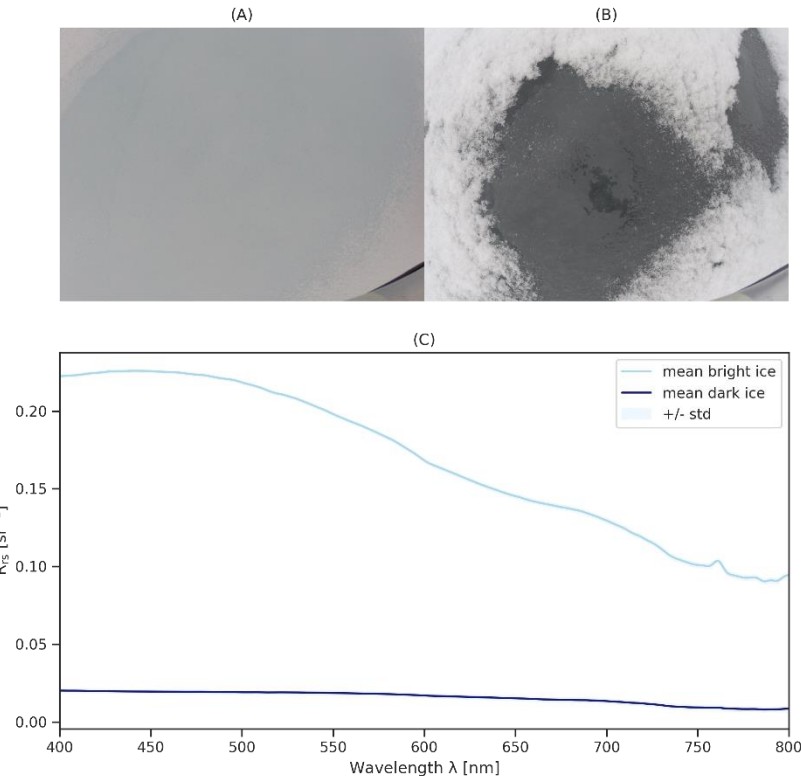

**Figure 1: Photos of bright (A) and dark (B) bare ice surfaces and respective reflectance spectra (C). We took the photos from approximately 50 cm (A) and 30 cm (B) above the surface.**

On 2 July 2017 between 22:28 UTC and 23:11 UTC, we performed twelve nadir measurements of a bare ice surface, likewise missing a surface scattering layer (Figure 2A), under clear sky conditions and a mean solar zenith angle of 74.89° with the Ibsen setup (Gege and König, 2019). Here, we use the average spectrum. The large standard deviation may be attributed to surface metamorphism during the measurement (Figure 2B).

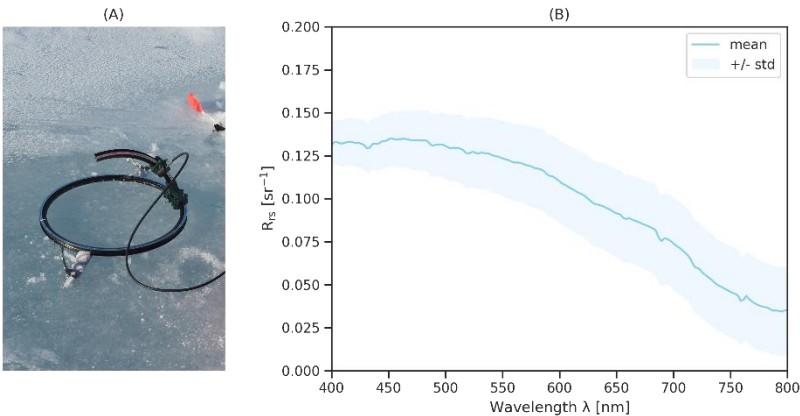

**Figure 2: Ibsen bare ice measurement setup (A). Spectra used in this study (B) were taken at nadir.**

### 2.1.2 Pond measurements

On 10 June 2017, we collected 49 melt pond spectra (Gege et al., 2019) and corresponding pond depths in three melt ponds. Two of the ponds had a bright blue color while the third one was very dark, which is also apparent in Figure 3. The pond site was located in a ridged area and ice thickness measurements from June 31, 2017 showed that ice thickness was $\geq 0.9$ m below the bright ponds and $\leq 0.5$ m below the dark pond indicating that the bright ice is older. We presume that the dark ice may have been a refrozen lead. However, no ice cores were analyzed to determine the respective ice types.

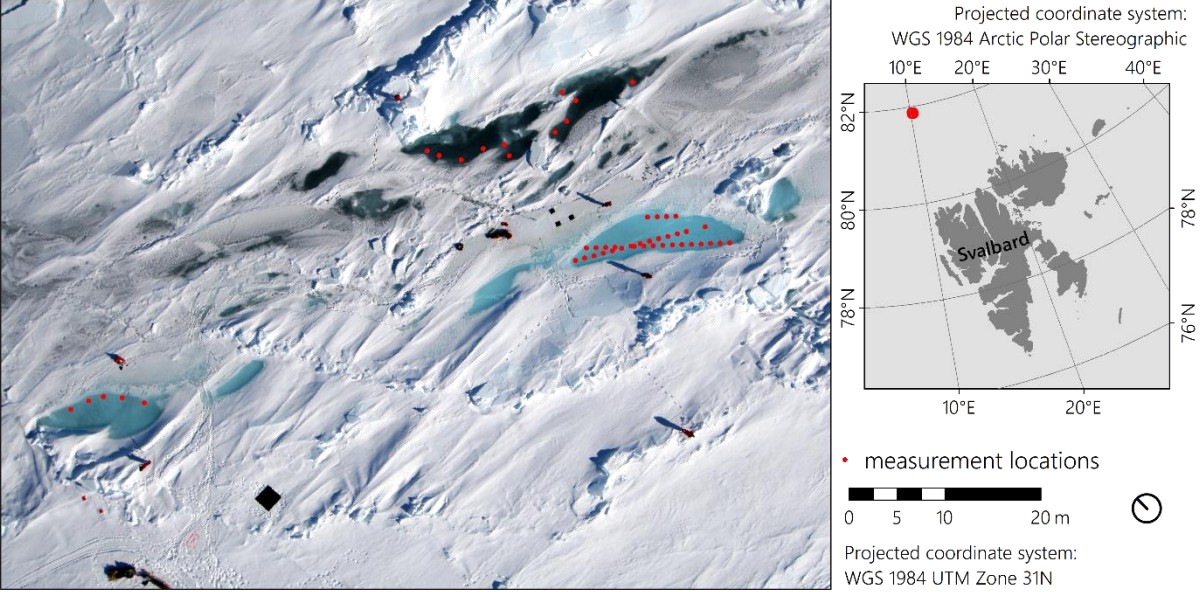

**Figure 3: Overview of measurement sites in the three ponds. Aerial photo: Gerit Birnbaum.**

Bottoms of the bright ponds were mostly smooth and solid but also featured few cracks and highly scattering areas that were very porous. The dark pond bottom was more heterogeneous and featured cracks and areas that were porous and riddled with holes (Figure 4).

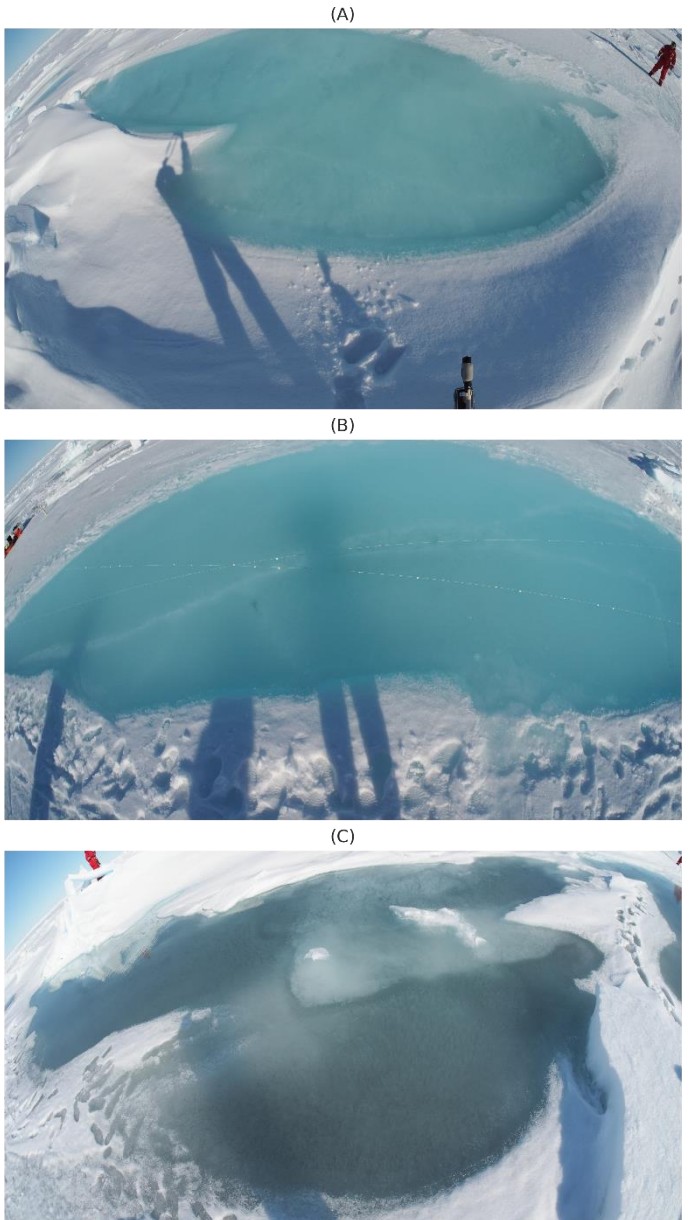

**Figure 4: Photos of the small (A) and large (B) bright and the dark pond (c). Photos: Peter Gege.**

At each pond, we referenced the Ocean Optics spectrometers using the Spectralon panel before data acquisition. We performed spectral measurements from the edge of the pond or waded through the pond avoiding shading. We did not observe any wind

induced disturbances of the water surface and waited for the water surface to settle before performing measurements inside the ponds. All measurements were performed under clear sky conditions between 12:23 local solar time and 14:43 local solar time, and corresponding solar zenith angles between 58.90° and 61.04°. Directly after each spectral measurement, we used a folding ruler to measure pond depth at the same location. Depths ranged between 6 cm and 25 cm with an average of 17.60 cm. Figure 5 illustrates the melt pond spectra and corresponding pond depths.

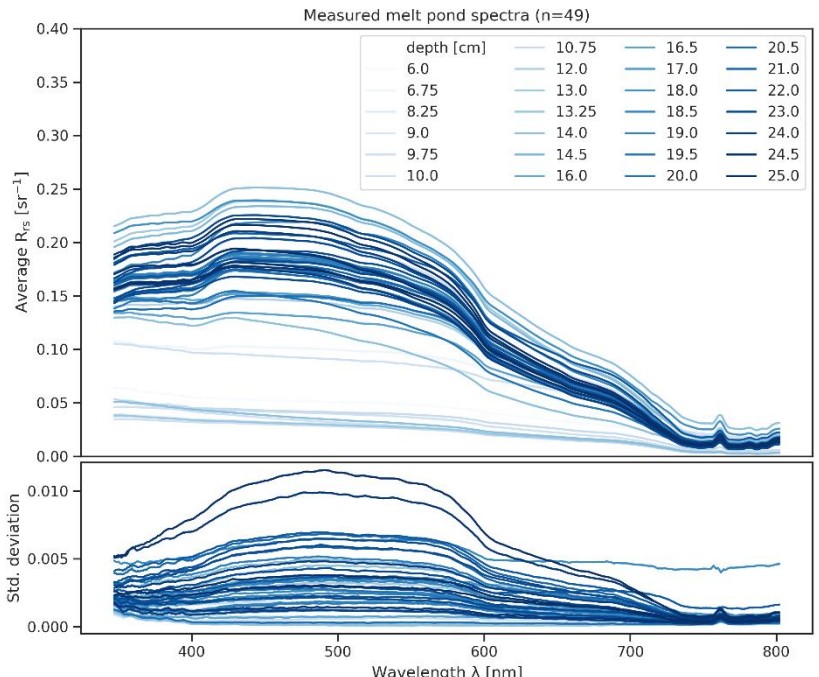


**Figure 5: Average reflectance spectra (top), standard deviations of 30 measurements (bottom) and corresponding pond depths.**

### 2.1.3 Data smoothing

Even though the spectra appear smooth at first view, the hardly visible amount of noise in the data becomes relevant for calculating derivatives. To smooth the spectra, we therefore resampled all spectra to a 1 nm spectral sampling by linear
interpolation, and then applied a running average filter with a width of 5 nm.

### 2.2 Model development

To develop an approach that does not require knowledge about on site ice characteristics, our model must be independent from changes of the bottom albedo, i.e. scattering characteristics of the underlying ice. It shall further be applicable to a wide range of pond depths up to 1.0 m. Because the in situ melt pond dataset is limited to shallow depths and biased towards bright blueish
ponds, we used the Water Color Simulator (WASI) to create a spectral library covering different bottom type mixtures and

depths. WASI is a software tool for the analysis and simulation of deep- and shallow-water spectra that bases on well-established analytical models (Gege, 2004, 2014, 2015; Gege and Albert, 2006). We used the forward mode of the program WASI-2D (v4.1) to generate libraries of melt pond spectra. The procedures are described in the following.

### 2.2.1 Simulated data

We used the Ocean Optics bare ice spectra from overcast sky conditions (Sect. 2.1.1 Ice spectra) as pond bottom reflectance. Analyses of optical properties of water samples showed only negligible amounts of chlorophyll-a, colored dissolved organic matter and total suspended matter. Moreover, Podgorny and Grenfell (1996) report that the signal of scattering in melt water is overwhelmed by the scattering in the bottom ice. We therefore defined a pure water column without additional absorbing or scattering water constituents and computed remote sensing reflectance in shallow water above the water surface according to

Eq. 2.20b in Gege (2015):

$$R_{rs}^{sh}(\lambda) = \frac{(1-\sigma)(1-\sigma_L^-)}{n_w^2} \cdot \frac{R_{rs}^{sh-}(\lambda)}{1-\rho_u \cdot Q \cdot R_{rs}^{sh-}(\lambda)} + R_{rs}^{surf}(\lambda) , \tag{4}$$

where $\sigma$, $\sigma_L^-$ and $\rho_u$ are the reflection factors for $E_d$ and upwelling radiance ($L_u^-$) and irradiance just below the water surface. $\sigma$ and $\rho_u$ are 0.03 and 0.54, respectively, while $\sigma_L^-$ is calculated from the viewing angle (0° for a nadir-directed sensor). $n_w$ is the refractive index of water ($\approx$ 1.33) and $Q$ is a measure of the anisotropy of the light field in water, approximated as 5 sr.

$R_{rs}^{sh-}$ is the remote sensing reflectance just below the water surface according to Albert and Mobley (2003):

$$R_{rs}^{sh-}(\lambda) = R_{rs}^-(\lambda) \cdot \left[1 - A_{rs,1} \cdot \exp\{-\left(K_d(\lambda) + k_{uW}(\lambda)\right) \cdot z_B\}\right] + A_{rs,2} \cdot R_{rs}^b(\lambda) \cdot \exp\{-K_d(\lambda) + k_{uB}(\lambda)) \cdot z_B\}, \tag{5}$$

where $A_{rs,1}$ and $A_{rs,2}$ are empirical constants, $K_d$, $k_{uW}$ and $k_{uB}$ describe the attenuation of the water body with depth $z_B$ defined by its absorption and backscattering, and the viewing and illumination geometry. The first part of Eq. (5) describes the contribution of the water body and the second part the contribution of the bottom. $R_{rs}^-$ is the remote sensing reflectance of deep

water just below the water surface defined by absorption and backscattering of the water body and the viewing and illumination geometry. $R_{rs}^b$ is the remote sensing reflectance of the bottom that is defined as the sum of the fractional radiances of all contributing bottom types defined by their albedos and under the assumption of isotropic reflection. $R_{rs}^{surf}$ in Eq. (4) is the ratio of radiance reflected by the water surface and $E_d$. We set $R_{rs}^{surf}$ to zero; thus, the last part of Eq. (4) can be ignored. We further used a solar zenith angle of 60°, similar to the in situ measurements, and a viewing angle of 0° (nadir).

We computed linear mixtures of the two measured bottom albedos in 25 % steps (100 % dark, 0 % bright; 75 % dark, 25 % bright; …; 0 % dark, 100 % bright). Using this setup, we generated a spectral look up table (LUT) by increasing pond depth from 0 to 100 cm in intervals of 1 cm, adequate for the great majority of melt ponds on Arctic sea ice. The final LUT contains 505 spectra (Figure 6).

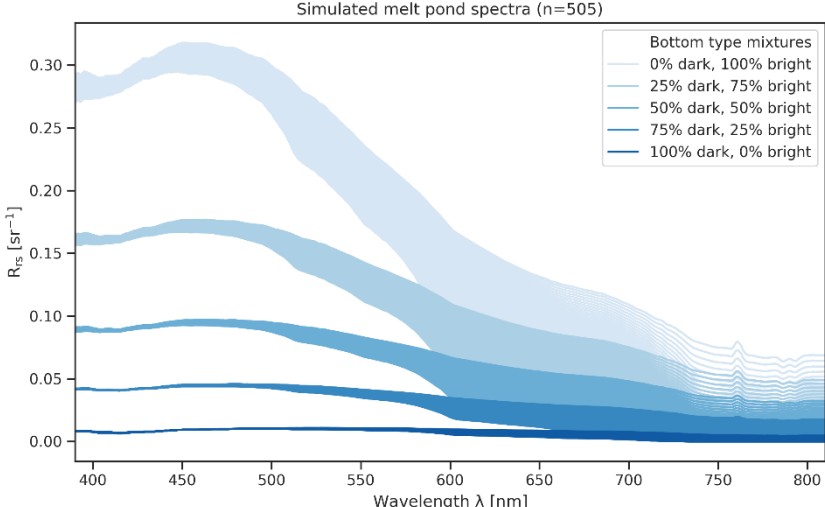

Figure 6: LUT generated with WASI-2D. Each of the five bottom type mixtures consists of 101 spectra (0 cm to 100 cm in 1 cm steps).

### 2.2.2 Data Processing

According to the Beer–Lambert law, the extinction of light at a certain wavelength in a medium is described by an exponential function. Here we assume that multiple scattering in melt water and (multiple) reflections at the pond surface, bottom and sidewalls can be neglected to approximate the radiative transfer. Figure 7A illustrates the exponential decrease of $R_{rs}$ with water depth at 700 nm for the five different bottom type mixtures. To linearize the effect, we computed the logarithm of the spectra (Figure 7B). Lastly, we computed the first derivative of the logarithmized spectra (Figure 7C) for each band by applying a Savitzky-Golay filter using a second order polynomial fit on a 9 nm window (The Scipy community, 2019b).

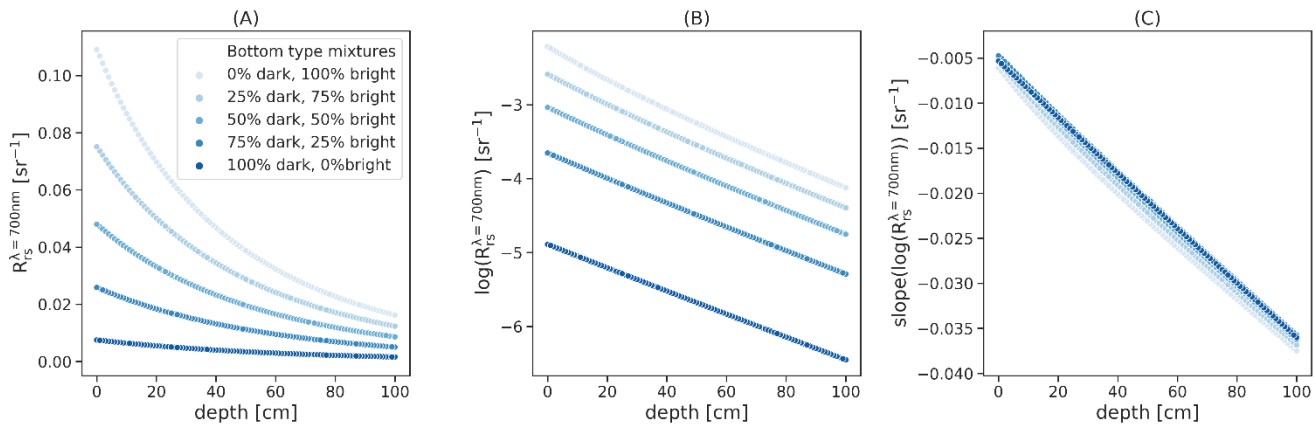

Figure 7: Processing of spectral data exemplified for $\lambda = 700$ nm.

We then computed Pearson's correlation coefficient ($r$) as (The Scipy community, 2019c):

$$r(x, y) = \frac{\sum_{i=0}^{n-1}(x_i - \acute{x})(y_i - \acute{y})}{\sqrt{\sum_{i=0}^{n-1}(x_i - \acute{x})^2 \sum_{i=0}^{n-1}(y_i - \acute{y})^2}},$$

(6)

where $x_i$ and $\acute{x}$ are the depth of the $i$-th sample and the average depth and $y_i$ and $\acute{y}$ are the slope of the logarithmized reflectance at a certain wavelength of the $i$-th sample and the average slope of the logarithmized reflectance at a certain wavelength; and $n$ is the number of samples.

The orange curve in Figure 8 illustrates the wavelength dependent correlation coefficients of the slope of the logarithmized spectra and pond depths in the LUT. We observe an almost perfect negative correlation in bands between 700 nm and 750 nm. We performed the same processing as for the simulated spectra for the in situ pond spectra. The blue curve in Figure 8 illustrates the wavelength dependent correlation coefficients of measured pond depth and the slope of the logarithmized in situ spectra. We likewise observe strong negative correlations in the wavelength region around 700 nm.

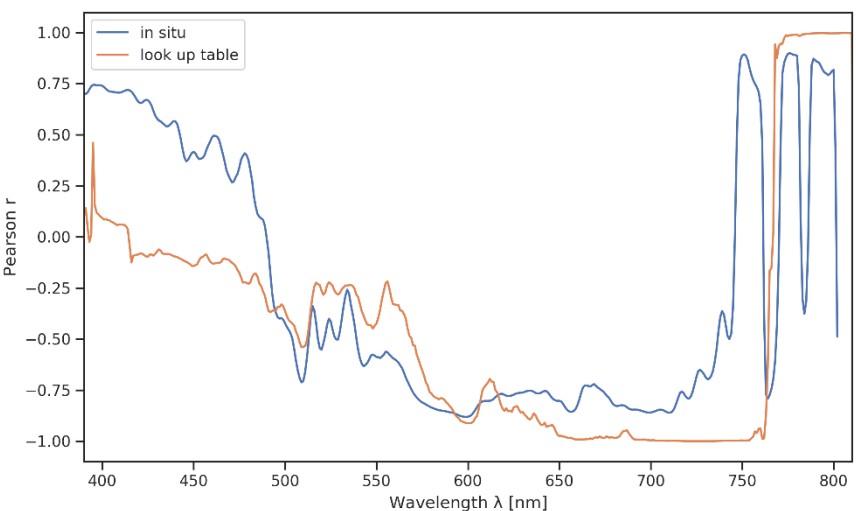

**Figure 8: Wavelength dependent correlation coefficients of pond depth with slope of log-scaled spectra for in situ measurements and simulated spectra.**

To investigate the similarity of the dark and bright ice spectra, we normalized both bottom spectra at 710 nm and found a high spectral similarity between ~ 590 nm and ~ 800 nm (Figure 9). Consequently, the slope of the logarithmized spectra is widely independent from the chosen bottom albedo in this wavelength region. Assuming that this also applies to ice spectra recorded under clear sky conditions, we used the Ibsen bare ice measurement to develop a model for clear sky conditions accordingly.

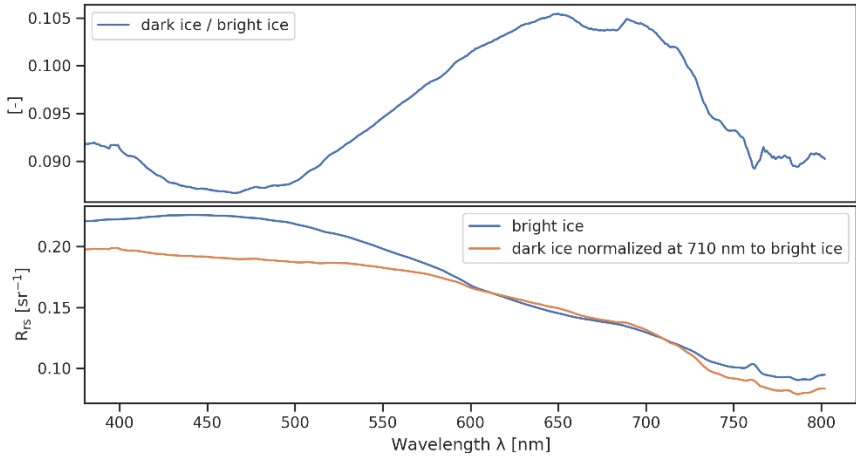

**Figure 9: Quotient of bright and dark bare ice spectra (top) and $R_{rs}$ of bright ice and dark ice normalized at 710 nm (bottom).**

### 2.2.3 Linear model

Due to the strong negative correlation in the simulated as well as in the measured data, we chose the slope of the logarithmized spectrum at 710 nm ($r$ = -1.0 and -0.86 for simulated and in situ data, respectively) to develop a simple linear model. We used

scikit-learn's Linear Regression function (Pedregosa et al., 2011) to fit a linear model to the simulated data with the Ibsen bare ice spectrum as bottom albedo using the method of Ordinary Least Squares.

We found that the solar zenith angle affects the slope and y-intercept of the linear model. Because the model shall be applicable to a wide range of solar zenith angles, we implemented a second model to derive slope and y-intercept of the linear model for various solar zenith angles. We used WASI to generate spectral libraries for different solar zenith angles (0°, 15°, 30°, 45°,

60°, 75°, 90°) and found that the resulting change of slope and y-intercept can each be described by an s-shaped curve. We used SciPy's optimize.curve_fit function (The Scipy community, 2019a) to fit generalized logistic functions (Richards, 1959) into the data. Using these functions, the model's slope and y-intercept can be computed for different solar zenith angles (Figure 10).

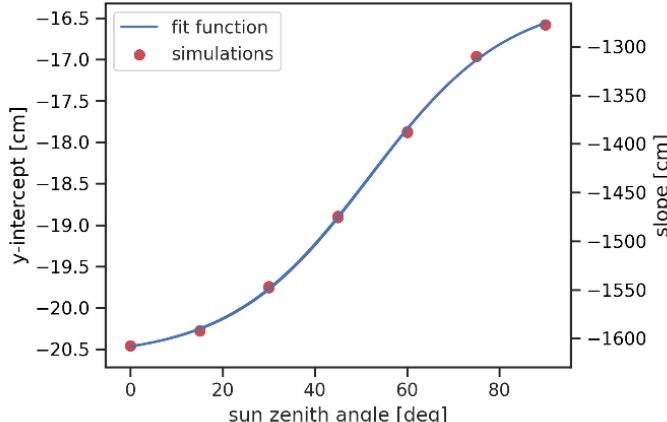

Figure 10: Change of model's y-intercept and slope with solar zenith angle. Generalized logistic function fit into the simulated data.

The model is

$$z = a(\theta_{sun}) + b(\theta_{sun}) \left[ \frac{\partial log\ R_{rs}(\lambda)}{\partial \lambda} \right]_{\lambda=710\ nm} \qquad (7)$$

where $z$ is the predicted pond depth and $\theta_{sun}$ is the solar zenith angle. $a$ and $b$ are offset and slope:

$$a(\theta_{sun}) = -20.6 + \frac{0.79}{0.8 + 5.8 \exp(-0.13 \cdot \theta_{sun})^{\frac{1}{2}}}\ [cm] \qquad (8)$$

and

$$b(\theta_{sun}) = -1619.8 + \frac{94743.64}{255.3 + 7855 \exp(-1.3 \cdot \theta_{sun})^{\frac{1}{19.9}}}\ [cm] \qquad (9)$$

We further computed the coefficient of determination ($R^2$) as recommended by Kvålseth (1985) as:

$$R^2(y, \hat{y}) = 1 - \frac{\sum_{i=0}^{n-1}(y_i - \hat{y}_i)^2}{\sum_{i=0}^{n-1}(y_i - \acute{y})^2}, \qquad (10)$$

where $y_i$ and $\hat{y}_i$ are the true (simulated) and predicted value of the $i$-th sample, $n$ is the number of samples and $\hat{y}_i = \frac{1}{n}\sum_{i=0}^{n-1} y_i$ (Pedregosa et al., 2011; scikit-learn developers, 2018). In addition, we also computed the root-mean-square error ($RMSE$) as:

$$RMSE(y, \hat{y}) = \sqrt{\frac{1}{n}\sum_{i=0}^{n-1}(y_i - \hat{y}_i)^2}, \qquad (11)$$

and the normalized $RMSE$ ($nRMSE$) as:

$$nRMSE(y, \hat{y}) = \frac{RMSE(y, \hat{y})}{\hat{y}} \cdot 100 \, , \tag{12}$$ For

the model described above we obtained a perfect correlation ($r = 1.0$; probability value ($p$) $= 8.9e^{-172}$), an $R^2$ of 1.0 and an

$RMSE$ of 0.56 cm ($nRMSE = 1\%$) on the simulated training data.

## 3 Results

We validated the model with the in situ melt pond dataset from dark and bright ponds (Sect. 2.1.2 Pond ) and observed a strong

linear and statistically significant correlation ($r = 0.86$; $p = 2.36e^{-15}$; $R^2 = 0.65$; $RMSE = 3.29$ cm and $nRMSE = 19\%$). Most

of the points scatter along the 1:1 line, except for one point where actual depth is 10 cm and predicted depth is 18 cm (Figure

11A). The externally studentized residual ($t$) (Kutner et al., 2004; Seabold and Perktold, 2010) classifies this point as an outlier

($t > 3$) and therefore we excluded this point from the data set. The removal of the outlier improves all performance measures

($r = 0.89$; $p = 4.34e^{-17}$; $R^2 = 0.68$; $RMSE = 3.11$ cm; $nRMSE = 18\%$). The slope of the line of best-fit increases to 0.9686 and

the intercept indicates an offset of 0.878 cm. If we further correct for the offset $R^2$ increases to 0.74 and $RMSE$ improves to

2.81 cm ($nRMSE = 16\%$). The blue line is the line of best fit between actual and predicted pond depths. The linear equation

of the line of best fit indicates that the model results in a small offset and a slope close to 1.0.

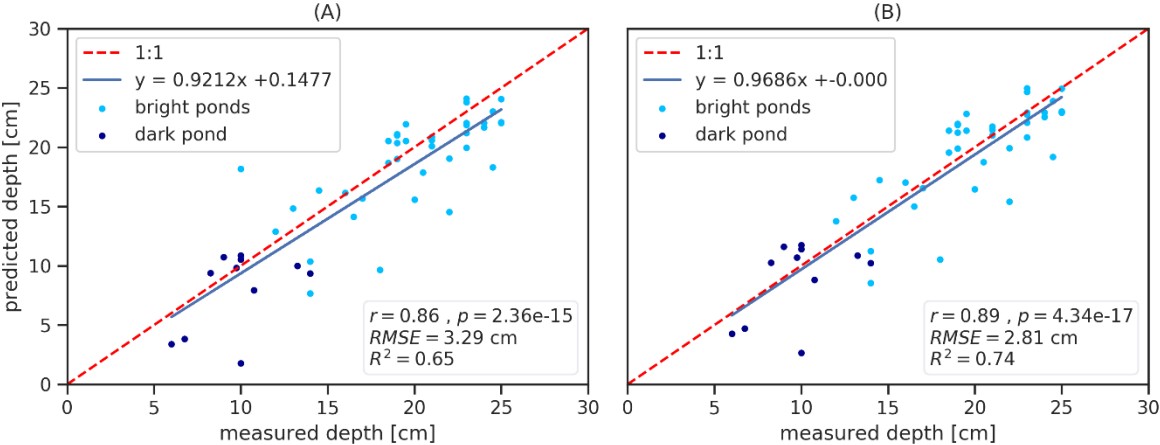

Figure 11: Measured vs. predicted depth for the entire dataset (A), with outlier removed and offset correction (B).

## 4 Discussion

Our results show that a simple model based on the derivative of the log-scaled $R_{rs}$ at 710 nm allows water depth retrieval of

dark and bright melt ponds on Arctic sea ice. The model training on simulated data and the independent testing using in situ

measurements prove the applicability of our approach.

### 4.1 Observational data

#### 4.1.1 Spectral measurements

Measurement of albedo have a long tradition in Arctic research (e.g. Grenfell, 2004; Nicolaus et al., 2010; Perovich, 2002; Perovich and Polashenski, 2012) because it is an important quantity in climate models and can be measured with a single irradiance detector. In this study, we conducted measurements of $R_{rs}$ because our model should be applicable to remote sensing data and the quantity measured in optical remote sensing is radiance. It is only appropriate to derive an accurate radiance directly from the albedo of a Lambertian surface. This assumption, however, is not valid for specular water surfaces and may easily introduce errors. Morassutti and Ledrew (1996) identified changing $E_d$ as the main error affecting reflectance data recording. To tackle this issue, we used a combination of two spectrometers described in Sect. 2.1 Observational data.

Field spectroscopy is influenced by external factors and the measurement design itself. In contrast to ruler measurements, the spectrometer acquires information of an area. To ease comparison and limit the influence of spatial heterogeneities, we used a fore optic with a 1° FOV to minimize the footprint (~ 1 cm at a height of 60 cm). However, holding the instruments perfectly still for a period of several seconds is challenging and even small changes in the position result in changes of the viewing angle, which increase the footprint of a measurement. For future campaigns, we therefore recommend using a gimbal to minimize the influence of roll and pitch of the hand-held spectrometer setup. Another issue might have been reflections of the black spectrometer housings at the water surface possibly contributing to the offset between modeled and measured data. Different refraction indices of wet and dry surfaces may cause part of the observed offset. Furthermore, using bottom albedos obtained from dry surfaces in WASI introduce a systematic offset. However, it remains unclear if the ice surface used to compute the spectral library was wet or dry.

Some of the scattering may be introduced by reflectances at the water surface, which we did not consider in the LUT computation because the necessary values for the parametrization are unknown. Another influence may be the different solar zenith angles between bare ice and pond measurements. The potential influence of the mentioned factors may be worth further examination to refine the model.

#### 4.1.2 Pond depth measurements

Measuring the depth of a pond may appear trivial but the bottom of a pond is frequently not flat and solid but can be slushy or riddled with holes. In addition, performing two measurements with a spectrometer and a folding ruler at the exact same location is difficult. We therefore recommend using a laser pointer at the end of the pole for orientation. These uncertainties explain some of the scattering in Figure 11. Interpretation of field photographs of the pond bottoms however did not indicate any systematic errors associated with pond bottom characteristics.

## 4.2 Model validity

The majority of the field data used in this study are from bright blue ponds (n=38) while fewer measurements were obtained in dark ponds (n=11). We addressed this limited diversity of field data by computing a comprehensive LUT. The model generates accurate results ($RMSE$ = 2.81 cm) on the entire in situ test data set and explains a large portion of its variability ($R^2$ = 0.74). On the data set from the dark pond $R^2$ is $< 0$ and $nRMSE$ = 35 %. The reason is that measurements from the dark pond are very shallow (6 – 14 cm) and, thus, relative errors are larger compared to the deeper bright ponds. In addition, the number of data points is very small and single outliers have a strong influence on performance metrics. The range of scattering around the 1:1 line (Figure 11) however is similar for the data from dark ($RMSE$ = 3.05 cm) and bright ponds ($RMSE$ = 2.49 cm), proving that the model's accuracy is similar for both subsets.

The data used in this study are the most comprehensive set of $R_{rs}$ and depth measurements from melt ponds on Arctic sea ice acquired under clear sky conditions. The data set, however, originates from only three ponds, covering a limited variability of bottom characteristics and pond depth. More validation data are desirable to explore the model capabilities to derive pond depth from deep dark and shallow bright ponds, for pond depth $> 25$ cm and for a wider range of bottom types and solar zenith angles. In addition, more tests are necessary to explore how the model performs when the assumptions formulated in Sect. 2.2 are violated, e.g. when algae, suspended matter or yellow substances are abundant in the pond water or in the ice below the pond.

We successfully developed a model to accurately derive the depth of melt ponds on Arctic sea ice without having to consider the bottom ice characteristics of the pond; yet, we assume that we cannot entirely avoid any influence. When fitting a model to the Ocean Optics LUT (Figure 7C), we observe scattering around the 1:1 line resulting in $RMSE$ of 1.88 cm ($nRMSE$ = 4 %). In the Ocean Optics LUT, however, the only variable parameter is bottom type mixture; we therefore conclude that the scattering results from the difference in bottom albedo. Consequently, bottom albedo may affect the model, which may explain some of the scattering in the test data.

Optical satellite data can only be obtained under clear sky conditions but remote sensing images are likewise acquired from helicopters and UAVs. These platforms also operate under diffuse illumination conditions, which are frequent in the Arctic. To check the validity of the model for overcast conditions, we applied the clear sky model to data from the same area acquired on 14 June 2017 during diffuse illumination conditions. The performance, however, is low (Figure 12) and shows a moderate correlation ($r$ = 0.64; $p$ = 2.6e$^{-4}$), an $R^2 < 0$ and an $RMSE$ of 12.76 cm ($nRMSE$ = 63 %). We attribute the low performance to the different illumination conditions. Under diffuse conditions a considerable part of the reflectance measured above the water surface is due to reflection of clouds at the water surface. Further, the optical path length of the incoming light in water changes under overcast conditions.

We therefore conclude that, the present model is only valid for clear sky conditions. The model accounts for the influence of varying solar zenith angles but field data was limited to solar zenith angles between 58.9° and 61°. To enlarge its validity range more field data covering different weather and illumination conditions is necessary.

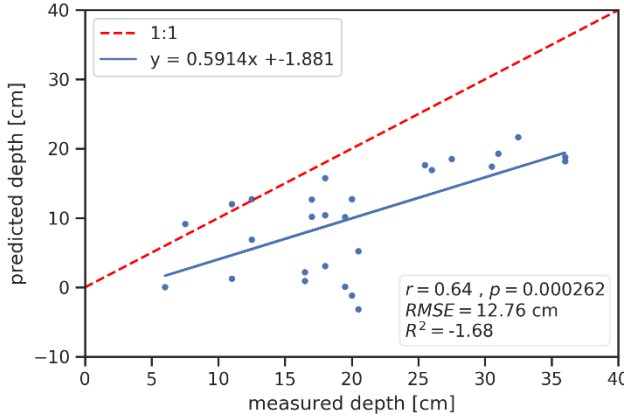

**Figure 12: Measured vs. predicted water depth for data acquired under overcast conditions on 14 June 2017.**

## 5 Conclusion

We present a linear model slope-based approach in the spectral region around 710 nm to retrieve the depth of melt ponds on Arctic sea ice. However, the model is not restricted to Arctic sea ice and may be tested in shallow supraglacial ponds as well. The model calibration on simulated data and independent validation on in situ data proves the applicability and robustness of our approach. The final model is valid for hyperspectral data ($R_{rs}$) acquired under clear sky conditions and addresses varying solar zenith angles.

We used WASI to generate a LUT of pond spectra for five different bottom albedos and pond depths between 0 and 100 cm assuming clear pond water. We found that the slope of the log-scaled $R_{rs}$ at 710 nm is widely independent from the bottom albedo and highly correlated with pond depth. Thus, we applied a linear model to retrieve pond depth from $R_{rs}$ in this wavelength region. Slope and y-intercept of the linear equation, however, change with solar zenith angle for which other models do not account for (e.g. Legleiter et al., 2014; Tedesco and Steiner, 2011). To overcome this limitation, we trained linear models for seven solar zenith angles between  and found that a general logistic function is able to describe the change of slope and y-intercept for each solar zenith angle. The inputs for our model therefore are the slope of the log-scaled $R_{rs}^{\lambda=710}$ and sun zenith angle. We successfully validated the model on in situ measurements ($r = 0.89$; $R^2 = 0.74$; $RMSE = 2.81$ cm; $nRMSE = 16$ %) with solar zenith angles between 58.9° and 61° and observed similar accuracies for bright and dark ponds.

The next step is the transfer to hyperspectral airborne and satellite systems, e.g. EnMAP (Guanter et al., 2016), to enable a synoptic view on the evolution of melt ponds on Arctic sea ice. One constraint may be the size of melt ponds, which requires a high spatial resolution. We further assume that the additive signals of the atmosphere and reflections of skylight at the water surface may complicate the retrieval of pond depth with remote sensors. In addition, the sensitivities and band settings of remote sensors also affect the transferability of our approach. Here, further testing and comprehensive ground truth data are

necessary. In these regards, we expect the Multidisciplinary drifting Observatory for the Study of Arctic Climate (MOSAiC) expedition to result in further improvements.

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

**Data availability**

The data used in this study are available at the PANGAEA data repository under doi.pangaea.de/10.1594/PANGAEA.908075.

**Author contribution**

MK and NO conceptualized the study. MK designed the methodology, curated and analysed the data, created and validated
the models, visualized results and wrote the original draft. NO critically reviewed the draft and both authors contributed in
editing and finalizing the paper.

**Competing interests**

The authors declare that they have no conflict of interest.

**Acknowledgements**

We thank Peter Gege for his encouragement and the provision of WASI. We highly appreciate the support by the German
Aerospace Center (DLR) Oberpfaffenhofen and especially thank Thomas Schwarzmaier, Stefan Plattner and Peter Gege for
the development and provision of the instruments used in this study. We further acknowledge the support of captain
Wunderlich, the crew and the chief scientists Andreas Macke and Hauke Flores of RV Polarstern cruise AWI_PS106_00 and
value the assistance provided by the colleagues supporting our fieldwork on PS106 especially Peter Gege, Gerit Birnbaum,
Niels Fuchs, Martin Hieronymi and Thomas Ruhtz. We would also like to thank Justin Mullins at Write About Science for his

valuable comments and Marcel Nicolaus for his estimation of the pond site's ice type situation. We acknowledge financial support by Land Schleswig-Holstein within the funding programme Open Access Publikationsfonds Finally, we thank two anonymous referees for their constructive critique, which helped us to improve the manuscript and Stef Lhermitte for his

editorial efforts.