# Peer review of "A linear model to derive melt pond depth on Arctic sea ice from hyperspectral data"

_The Cryosphere, 2019_

## Referee Comment (RC1) · Anonymous Referee #1 · 7 Jan 2020

This manuscript describes a model for the retrieval of melt pond depth from melt pond reflectance at 710 nm wavelength. The model was validated using numerous spectra / depth measurements from three unique melt ponds at a single location on a single day.

The manuscript is easy to read, mostly clearly written (see minor points below), and of general interest to the sea ice remote sensing community.

The authors make the assumption that the reflectance spectra of light and dark bare ice can serve as proxies for the reflectance spectra of the pond floors of light and dark ponds. I think this likely not true. Melting bare ice typically develops a surface scattering layer, which is likely not present where the ice is ponded. If I follow the arguments made by the authors later in the manuscript, it may turn out that their model

is very insensitive to this assumption, which is good. But, I do think the assumption merits some discussion when it is first presented (section 2.1.1).

It appears that there are 49 data points used in the validation of this model. That sounds like a large number, but I am concerned that they all come from only 3 distinct ponds. It is not stated whether the site was first-year or multiyear ice. Presumably, unless the pond floors were rafted ice, the optical properties of the pond floors for each pond were likely homogeneous? I suspect there is considerable variability in pond floor properties beyond what was sampled in these three ponds.

I wonder if it would be useful to compare the spectra shown in Fig. 5 with the spectra shown in Fig. 4 of Light et al. (2015; https://doi.org/10.1002/2015JC011163)? Eyeball comparison suggests the albedos in that study are spectrally flatter than the reflectance spectra shown here.

In the discussion (line 206) the authors declare the "universality" of this approach. I would argue that data from 3 melt ponds (all observed on the same day) likely does not show convincing universality! Also, the fact that the model is only valid for solar angles between 58.9 and 61 degrees makes it not truly universal.

Minor points: I think it would be helpful to include "sea ice" in the title. This study is specific to ponds on sea ice, and it may not be applicable to other types of ponds (e.g., on glaciers, ice shelves).

Line 7: "vertical melt pond evolution" is not clear. Do the authors mean "melt pond deepening"?

Line 10: "slope of the log-scaled remote sensing reflectance..." Isn't it really the "slope of the log-scaled reflectance as a function of depth"?

Line 24: It is not clear what is meant here by "open"? Do the authors mean "ponds with no ice at their surface"?

Line 49 (and other places as well, line 55, etc.): sampling rate < 1 nm? The unit "nm"

is a length, not a rate?

Line 54: "...within the scope of a goniometer experiment"– OK, so goniometer is a tool used in that experiment, but it would be helpful to give that experiment a more general name– maybe "...within the scope of an angle-resolving BRDF experiment"?

Line 69: "... negligible standard deviation" of what?

Line 85: Please give the time of the measurements in local solar time. I understand UTC is useful for syncing across datasets, but local solar time is essential for interpreting optical measurements.

Line 96: "expert knowledge" would help if this could be more specific.

Line 102: "spectral libraries of melt pond spectra" redundant wording

line 107: this means there are no constituents contributing to the absorption or scattering, but the absorption of the pure water is still accounted for?

Line 131: It should be noted that the Beer-Lamber law applies strictly to media that have no multiple scattering, in one-dimensional domains, in regions not affected by boundary conditions. The ponds in this study should be good candidates for the application of this law, but strictly speaking, it is an approximation to a full radiative transfer treatment.

Figure 6: The relationships shown here are just confirmation that the model used (Eqs 4, 5) satisfy Beer's Law, yes?

Line 239 (numerous other instances throughout manuscript): data "are" not "is"

line 245: "widely independent"? This needs to be clarified. Is "independent" sufficient?

---

## Author Comment (AC1) · 10 Jan 2020

Dear Anonymous Referee 1,

thanks a lot for your thorough evaluation and helpful comments on our manuscript. Please find our answers to your questions and our suggestions for integration of your improvements in the following paragraphs. For reasons of clarity referee comments are italic and answers are blue.

Best regards
Marcel König
* * *
*The authors make the assumption that the reflectance spectra of light and dark bare*

[Figure]

*ice can serve as proxies for the reflectance spectra of the pond floors of light and dark ponds. I think this likely not true. Melting bare ice typically develops a surface scattering layer, which is likely not present where the ice is ponded. If I follow the arguments made by the authors later in the manuscript, it may turn out that their model is very insensitive to this assumption, which is good. But, I do think the assumption merits some discussion when it is first presented (section 2.1.1).*

In fact, the bare ice surfaces did not consist of the typical surface scattering layer (compare Figure 1 A and B). We are going to add this information to Section 2.1.1.
* * *
*It appears that there are 49 data points used in the validation of this model. That sounds like a large number, but I am concerned that they all come from only 3 distinct ponds. It is not stated whether the site was first-year or multiyear ice. Presumably, unless the pond floors were rafted ice, the optical properties of the pond floors for each pond were likely homogeneous? I suspect there is considerable variability in pond floor properties beyond what was sampled in these three ponds.*

Unfortunately, we did not collect ice cores from the test site to measure salinity profiles. Yet, ice thickness under the bright ponds was $\geq$ 0.9 m while it was $\leq$ 0.5 m below the dark ponds. From the patterns of nearby ridges, we presume that the bright ponds were located on old ice – possibly second or multiyear ice – and that the dark ponds were located on a refrozen lead. We can add this information to Section 2.
While the bottom texture was similar in the bright ponds, variability was higher in the dark pond. Line 232 is pointing to the differences in texture. However, you are right that the dataset covers only a portion of the possible variability in bottom texture and optical properties. We are going to add a sentence to Section 4.2.
* * *
*I wonder if it would be useful to compare the spectra shown in Fig. 5 with the spectra shown in Fig. 4 of Light et al. (2015; https://doi.org/10.1002/2015JC011163)? Eyeball comparison suggests the albedos in that study are spectrally flatter than the reflectance*

*spectra shown here.*

We also thought about using spectra from other studies. Yet, we are afraid that the different measurement techniques and corresponding units (irradiance vs. remote sensing reflectance) hamper a direct comparison.
* * *
*In the discussion (line 206) the authors declare the "universality" of this approach. I would argue that data from 3 melt ponds (all observed on the same day) likely does not show convincing universality! Also, the fact that the model is only valid for solar angles between 58.9 and 61 degrees makes it not truly universal.*

You are right "universality" is probably a little overstated. We are going to change it to "applicability". The model has been tested on solar angles between 58.9 and 61 degrees but was calibrated for zenith angles between 0 and 90 degrees.
* * *
*Minor points: I think it would be helpful to include "sea ice" in the title. This study is specific to ponds on sea ice, and it may not be applicable to other types of ponds (e.g., on glaciers, ice shelves).*

Thank you for this valid comment. We suggest changing the title to "A linear model to derive melt pond depth on Arctic sea ice from hyperspectral data".
* * *
*Line 7: "vertical melt pond evolution" is not clear. Do the authors mean "melt pond deepening"?*

Yes. We are going to use "pond deepening" instead.
* * *
*Line 10: "slope of the log-scaled remote sensing reflectance..." Isn't it really the "slope of the log-scaled reflectance as a function of depth"?*

You are right. We are going to change it to "slope of the log-scaled reflectance at 710 nm as a function of depth".

*Line 24: It is not clear what is meant here by "open"? Do the authors mean "ponds with no ice at their surface"?*

Yes. You are right that open may also be understood as to melt through. Although "The reflected optical signal from melt ponds with ice-free surface . . ." may be the more accurate formulation.

*Line 49 (and other places as well, line 55, etc.): sampling rate < 1 nm? The unit "nm" is a length, not a rate?*

We are going to replace "rate" by "interval".

*Line 54: "...within the scope of a goniometer experiment"– OK, so goniometer is a tool used in that experiment, but it would be helpful to give that experiment a more general name– maybe "...within the scope of an angle-resolving BRDF experiment"?*

We will change it as suggested.

*Line 69: ". . . negligible standard deviation" of what?*

The standard deviation of the 30 spectra of one measurement. We are going to add this information.

*Line 85: Please give the time of the measurements in local solar time. I understand UTC is useful for syncing across datasets, but local solar time is essential for interpreting optical measurements.*

We will change all time designations to local time.

*Line 96: "expert knowledge" would help if this could be more specific.*

We suggest "... that does not require knowledge about specific ice characteristics".
* * *
*Line 102: "spectral libraries of melt pond spectra" redundant wording*

Right. We are going to change it to "libraries of melt pond spectra".
* * *
*line 107: this means there are no constituents contributing to the absorption or scattering, but the absorption of the pure water is still accounted for?*

Yes.
* * *
*Line 131: It should be noted that the Beer-Lamber law applies strictly to media that have no multiple scattering, in one-dimensional domains, in regions not affected by boundary conditions. The ponds in this study should be good candidates for the application of this law, but strictly speaking, it is an approximation to a full radiative transfer treatment.*

We suggest adding the following statement to the paragraph: "Here we assume that multiple scattering in melt water and (multiple) reflections at the pond surface, bottom and sidewalls can be neglected to approximate the radiative transfer."
* * *
*Figure 6: The relationships shown here are just confirmation that the model used (Eqs 4, 5) satisfy Beer's Law, yes?*

Yes, but they also visualize the data processing and illustrate the linear correlation (C).
* * *
*Line 239 (numerous other instances throughout manuscript): data "are" not "is"*

Thanks, we are going to change this.
* * *
*line 245: "widely independent"? This needs to be clarified. Is "independent" sufficient?*

Due to the RMSE of 1.88 cm when fitting the model to the Ocean Optics spectral library, we believe that an influence cannot be entirely avoided (compare lines 245ff).

---

## Referee Comment (RC2) · Anonymous Referee #2 · 2 Mar 2020

König and Oppelt developed a model to retrieve the depth of melt ponds on sea ice using in situ spectrometer measurements. The manuscript is structured logically, generally well presented and the methods appear sound. The purpose of the work, however, is not clearly articulated, many of the concluding statements are not supported by the evidence, and the novelty and significance of this work is limited. For these reasons, I recommend that the authors either substantially broaden the scope of the manuscript or consider submitting to a more specialized journal. I outline some potential improvements in my major and specific comments below.

Major comments

The authors provide no justification for deriving melt pond depths in the introduction. Do deeper ponds have a substantial impact on sea ice energy balance? Are melt pond

volumes required inputs for sea ice modeling? From the introduction, it appears we only need melt pond fraction for forecasting September sea ice area. Without these statements, it is difficult to understand why the authors have put effort into deriving a model for simulating melt pond depths.

The conclusions of this study are not supported by the evidence. In L204-205, the authors claim that their approach is "universal" and able to derive depths from dark and bright melt ponds. However, in the results, it looks like the R2 for the dark ponds is much worse than for bright ponds. It is misleading to claim that their model is accurate for dark ponds.

Likewise, the authors state that their study provides "the most comprehensive set of Rrs and depth measurements from Arctic melt ponds...". However, the study only presents 49 depths from three melt ponds with a depth range of 6 to 25 cm. A quick search of the literature reveals that this statement misleading. Malinka et al. (2018) https://doi.org/10.5194/tc-12-1921-2018 used coincident depth and spectra measurements from three different areas of the Arctic (SHEBA experiment in 1998, Barrow in 2008 and the Polarstern in 2012). Tedesco and Steiner (2011) doi:10.5194/tc-5-445-2011 and Legleiter et al. (2014) doi:10.5194/tc-8-215-2014 collected hundreds if not thousands of coincident depth and spectra measurements in a melt pond on the Greenland Ice Sheet that had depths of up to 10 m. The authors should review this literature (including those studies from the Greenland Ice Sheet) before making such claims.

The problems outlined above raise questions about the novelty and significance of the study. The only real result in the abstract is that the "...results indicate that pond depth is retrievable from optical data under clear sky conditions". As far as I understand (and based on the references in the previous paragraph) this is not a novel finding. The authors should think more deeply about how their study advances our understanding and build on previous research. However, in the present version, this study may only have limited interest to the cryospheric community.

The writing style is vague in many places. For example: L5: "Key elements" L33: "easy-to-use" L96: "expert knowledge" L234: "hitting the same spot" L235: "tricky" L245: "widely independent" The authors should consider being more specific where possible to improve the readability.

Specific comments

L28-209: Considering the apparent similarity between this work and Malinka et al. (2018), the authors should consider adding a much more thorough description of how the two models differ in the introduction.

L29: R2 of 0.62... against what? In situ depth measurements?

L52: How long was the pole?

L57: Replace "has been" with "was"

L67: Why was the ice surface dark?

P126: Please justify why pond depth is extrapolated to 1 m when the maximum pond depths in this study were 25 cm.

L186: Excessive referencing of Pedregosa et al. (2011) and scikit-learn developers is unnecessary. Consider removing some of these references.

L191: Please separate dark and bright ponds when presenting correlation coefficients.

L194: "18.17" check precision

Figure 1: Scale bar required for images

Figure 2A: I thought the spectrometer was mounted on a pole. Why is it on the ice surface?

---

## Author Comment (AC2) · 13 Mar 2020

Dear Anonymous Referee 2,

thanks a lot for your evaluation of our manuscript. Your valuable comments were a big help. We tried to improve the manuscript accordingly. Please find our answers to your questions and our suggestions for integration of your improvements in the following paragraphs. For reasons of clarity referee comments are italic and answers are blue.

Best regards, Marcel König
* * *
*The authors provide no justification for deriving melt pond depths in the introduction.*

[Figure]

*Do deeper ponds have a substantial impact on sea ice energy balance? Are melt pond volumes required inputs for sea ice modeling? From the introduction, it appears we only need melt pond fraction for forecasting September sea ice area. Without these statements, it is difficult to understand why the authors have put effort into deriving a model for simulating melt pond depths.*

Thanks a lot asking these questions. Melt pond depth is a parameter in the Los Alamos sea ice model CICE (Flocco et al., 2012 [doi: 10.1029/2012JC008195]; Hunke et al., 2013 [doi: 10.1016/j.ocemod.2012.11.008]) and the ECHAM5 general simulation model (Pedersen et al., 2009 [doi: 10.1029/2008JD010440]). Po­lashenski, Perovich and Courville (2012, doi: 10.1029/2011JC007231) showed that deriving pond depth from area fraction is impractical. Lecomte et al. (2011 [doi: 10.3189/172756411795931453], 2015 [doi: 10.1016/j.ocemod.2014.12.003]) use pond depth to parameterize melt pond albedo. Palmer et al. (2014 [doi: 10.1016/j.dsr2.2014.03.016]) use melt pond depth to model primary production below the sea ice. Lee et al. (2012) [doi: 10.1029/2011JC007717] call for more measurements to increase our understanding of how melt ponds on sea ice change in the context of climate change. We will integrate this information into the introduction of our manuscript.
* * *
*The conclusions of this study are not supported by the evidence. In L204-205, the authors claim that their approach is "universal" and able to derive depths from dark and bright melt ponds. However, in the results, it looks like the R2 for the dark ponds is much worse than for bright ponds. It is misleading to claim that their model is accurate for dark ponds.*

We agree that "universal" is overstated and suggest changing "universality" in L206 to "applicability". You are right that R2 is worse for the data subset from the dark pond. The problem here is that measurements from the dark pond are few (n=11) and

shallow (6 cm - 14 cm). Thus, measurement inaccuracies and outliers have a large impact on r and R2. For a precise assessment, we need more data from deep, dark ponds. The same is true for bright ponds: We need more data from shallow bright ponds and more data from ponds deeper than 25 cm in general. However, Figure 10 illustrates that measurements from the dark pond and bright ponds scatter well around the 1:1 line, suggesting that the model's accuracy is similar for dark and bright ponds. This is supported by their similar RMSEs (3.04 cm for the dark pond and 2.49 cm for the bright pond, respectively). We therefore suggest adding a few lines to the discussion section to address these issues.
* * *
*Likewise, the authors state that their study provides "the most comprehensive set of Rrs and depth measurements from Arctic melt ponds...". However, the study only presents 49 depths from three melt ponds with a depth range of 6 to 25 cm. A quick search of the literature reveals that this statement misleading. Malinka et al. (2018) https://doi.org/10.5194/tc-12-1921-2018 used coincident depth and spectra measurements from three different areas of the Arctic (SHEBA experiment in 1998, Barrow in2008 and the Polarstern in 2012). Tedesco and Steiner (2011) doi:10.5194/tc-5-445-2011 and Legleiter et al. (2014) doi:10.5194/tc-8-215-2014 collected hundreds if not thousands of coincident depth and spectra measurements in a melt pond on the Greenland Ice Sheet that had depths of up to 10 m. The authors should review this literature (including those studies from the Greenland Ice Sheet) before making such claims.*

You are right that a huge data set of spectral measurements exists. However, the data sets mentioned in Malinka et al. 2018 are in units of albedo not Rrs. Our reasons to measure Rrs instead of albedo are explained in Section 4.1.1. Melt ponds on ice sheets and sea ice differ substantially due to the optical thickness of the bottom ice and maximum depth. While melt ponds on ice sheets can reach depths of tens of meters, ponds on sea ice are mostly shallower than 1 m. More importantly, pond bottoms

on sea ice may be optically thin, i.e. ice thickness influences the reflectance signal, whereas this is not the case in glacial ponds. As suggested by Anonymous Referee 1, we will include "sea ice" into the title to avoid confusions with melt ponds on glaciers.
* * *
*The problems outlined above raise questions about the novelty and significance of the study. The only real result in the abstract is that the "...results indicate that pond depth is retrievable from optical data under clear sky conditions". As far as I understand (and based on the references in the previous paragraph) this is not a novel finding. The authors should think more deeply about how their study advances our understanding and build on previous research. However, in the present version, this study may only have limited interest to the cryospheric community.*

Thanks for your thorough review. We assume that the model presented by Malinka et al. struggles to retrieve melt pond depth because different combinations of ice scattering coefficient, ice thickness and pond depth may result in ambiguous spectra. Therefore, we propagate to use information from specific wavelength regions influenced primarily by the thickness of the melt water layer for the retrieval of pond depth. We will emphasize this in the updated version of the manuscript.
* * *
*The writing style is vague in many places. For example: L5: "Key elements" L33:"easy-to-use" L96: "expert knowledge" L234: "hitting the same spot" L235: "tricky"L245: "widely independent" The authors should consider being more specific where possible to improve the readability.*

L5: We do not understand the problem with "key elements".
L33: We suggest changing "easy-to-use" to "easy to apply".
L96: We will change "expert knowledge" to "knowledge about local conditions".
L234f: According to your comments, we will change the sentence to: "In addition, performing two measurements with a spectrometer and folding ruler at the exact same

location is difficult."
L245: "widely independent" is explained in the following lines of the paragraph.
* * *
*Specific comments*

*L28-209: Considering the apparent similarity between this work and Malinka et al. (2018), the authors should consider adding a much more thorough description of how the two models differ in the introduction.*

See comment above.
* * *
*L29: R2 of 0.62...against what? In situ depth measurements?*

Yes. We suggest changing the sentence to: "... resulting in an R2 of 0.62 (N=26) against in situ pond depths between 6 cm and 50 cm ... "
* * *
*L52: How long was the pole?*

The pole had a length of 1 m. We will add this information.
* * *
*L57: Replace "has been" with "was"*

We will replace "has been" with "was".
* * *
*L67: Why was the ice surface dark?*

The color and brightness of an ice surface is governed by the scattering characteristics of the ice, which in turn are defined by its physical properties, e.g. bubbles, structure, thickness etc. However, we do not have any measurements that may be

used to assess this information.

*P126: Please justify why pond depth is extrapolated to 1 m when the maximum pond depths in this study were 25 cm.*

We suggest adding the following: "... in intervals of 1 cm, adequate for the great majority of melt ponds on Arctic sea ice."

*L186: Excessive referencing of Pedregosa et al. (2011) and scikit-learn developers is unnecessary. Consider removing some of these references.*

We will remove the second reference in L187.

*L191: Please separate dark and bright ponds when presenting correlation coefficients.*

As described above we will rewrite the discussion section and integrate separate performance measures for bright and dark ponds.

*L194: "18.17" check precision*

We will change "18.17" to "18".

*Figure 1: Scale bar required for images*

Adding a scale bar after taking the images is unfortunately not possible. Diameters of the bare ice surfaces are approximately 0.5 m (A) and 0.3 m (B). We can include this information in the caption.

*Figure 2A: I thought the spectrometer was mounted on a pole. Why is it on the*

*icesurface?*

Only the Ocean Optics setup was mounted on a pole. Figure 2A shows the Ibsen setup mounted on the field goniometer described in Section 2.1.

---

## Author Response (AR1)

Dear Stef Lhermitte and anonymous referees,

Thank you very much again for your thorough and very constructive review of our manuscript. We tried to address all the issues and questions that came up during the review process and revised the manuscript accordingly. We are confident that your comments improved the manuscript's quality a lot. Thank you for this productive review process.

First, we would like to apologize for the brevity of some of our comments in the interactive discussion. It is the first time that we undergo this form of review process and we expected the review to be similar to a chat.

As advised, we revised our manuscript based on the comments related i) to similar observation/models for melt ponds on grounded ice and ii) to the novelty.

In our revised version, we restructured the Introduction and now refer to the work of Legleiter et al. (2014) and Tedesco and Steiner (2011) on supraglacial lakes on the Greenland ice sheet. We describe their work and reason why both models are unsuitable for melt ponds on sea ice. We also describe that the great challenge in both studies is the wide range of depth in supraglacial ponds. As suggested by Referee #2 we further specified the description of the models of Lu et al. (2016) and Malinka et al. (2018) and formulated an educated guess about the reasons for their problems in pond depth retrieval. Please read our restructured Introduction for further details.

Thanks to your constructive review and editorial, we highlighted the important findings that underline the novelty of our model (see Abstract and Conclusion). (i) Our model uses near infrared light for the retrieval of melt pond bathymetry on Arctic sea ice with unprecedented accuracy and without having to consider bottom albedo. (ii) It addresses the influence of changing sun zenith angle and (iii) is of semi-empirical nature due to the model training on simulated data. This has the advantage that our model - in contrast to purely empirical models such as the one described in Legleiter et al. (2014) - is unbiased towards a certain range of depth or bottom albedo.

Although the model is fitted for ponds on sea ice, it would be worth testing it on shallow pond measurements from supraglacial lakes. We added this idea to the discussion section. However, we believe that this analysis needs to be done in a separate study.

We further added information about the difference between Rrs and R to the discussion section. As Stef correctly pointed out albedo may be used as a proxy for satellite measurements when assuming lambertian behavior. However, this assumption is not complied over water surfaces, which do not reflect lambertian but are characterized by Fresnel reflection. A direct comparison between Rrs and R measurements over water is therefore not trivial. In addition, satellite and airborne remote sensors acquire radiance and not irradiance. For this reason, Rrs has replaced albedo in remote sensing and we chose it for our measurements accordingly.

In the following tables, we reply to the referee comments in more detail.

We are confident that our revised version fulfils the requirements for publication in The Cryosphere.

Best regards,

Marcel König and Natascha Oppelt

**Referee #1**

**Major comments**

| Comment | Answer |
|---|---|
| The authors make the assumption that the reflectance spectra of light and dark bare | We chose the bare ice surfaces because we did not observe a surface scattering layer (compare |

| | |
|---|---|
| ice can serve as proxies for the reflectance spectra of the pond floors of light and dark ponds. I think this likely not true. Melting bare ice typically develops a surface scattering layer, which is likely not present where the ice is ponded. If I follow the arguments made by the authors later in the manuscript, it may turn out that their model is very insensitive to this assumption, which is good. But, I do think the assumption merits some discussion when it is first presented (Section 2.1.1). | Figure 1A,B) and made the assumption the corresponding spectra serve as good candidates for pond bottom spectra. We added this information as well as our assumption to Section 2.1.1. |
| It appears that there are 49 data points used in the validation of this model. That sounds like a large number, but I am concerned that they all come from only 3 distinct ponds. It is not stated whether the site was first-year or multiyear ice. Presumably, unless the pond floors were rafted ice, the optical properties of the pond floors for each pond were likely homogeneous? I suspect there is considerable variability in pond floor properties beyond what was sampled in these three ponds. | As we mentioned in our first reply, no ice cores were analyzed to define the ice types below the ponds. Ice thickness measurements from June 31 however indicate that the bright ice is older (possibly second year or multiyear ice) than the dark one (possibly a refrozen lead). We added these details to Section 2.1.2. Regarding the bottom texture of the ponds, we likewise added a description and field photographs to Section 2.1.2. We also added a paragraph to the discussion in Section 4.2 pointing towards the limited amount of variability within in three ponds and highlighted the need for more data. |
| I wonder if it would be useful to compare the spectra shown in Fig. 5 with the spectra shown in Fig. 5 of Light et al. (2015; https://doi.org/10.1002/2015JC011163)? Eyeball comparison suggests the albedos in that study are spectrally flatter than the reflectance spectra shown here. | Of course, we were aware of the large database of albedo spectra from previous studies. Yet, as we pointed out above, a comparison of albedo and Rrs measurements is not trivial due to the non-lambertian characteristics of the water surface. We refer to this issue in Section 4.1.1. |
| In the discussion (line 206) the authors declare the "universality" of this approach. I would argue that data from 3 melt ponds (all observed on the same day) likely does not show convincing universality! Also, the fact that the model is only valid for solar angles between 58.9 and 61 degrees makes it not truly universal. | We agree that "universal" was overstated and therefore used the term "applicability" instead. You are right that the limited amount of spectra as well as the variability of depth and bottom characteristics in the test data requires more field data for validation. We restructured the discussion accordingly (Section 4.2). We also agree that more testing on different sun zenith angles is desirable. The model, however, is valid for all sun zenith angles. In comparison to purely empirical models trained on field measurements only, our novel methodology based on a simulated spectral library has the virtue that variations due to sun zenith angle are accounted for by the model (compare Section 2.2.3). |

**Minor comments**

We addressed most of the minor comments according to your suggestions and as described in the first response. Only regarding your last comment, we came up with a new formulation.

| line 245: "widely independent"? This needs to be clarified. Is "independent" sufficient? | We agree that the formulation was not precise and changed the sentence to: "We successfully developed a model to accurately derive the depth of melt ponds on Arctic sea ice without having to consider the bottom ice characteristics of the pond." |
|---|---|

**Referee #2**

**Major comments**

| The authors provide no justification for deriving melt pond depths in the introduction. Do deeper ponds have a substantial impact on sea ice energy balance? Are melt pond volumes required inputs for sea ice modeling? From the introduction, it appears we only need melt pond fraction for forecasting September sea ice area. Without these statements, it is difficult to understand why the authors have put effort into deriving a model for simulating melt pond depths. | You are right that the justification was too sparse in the previous version of the manuscript. We therefore included some references in the revised introduction. |
|---|---|
| The conclusions of this study are not supported by the evidence. In L204-205, the authors claim that their approach is "universal" and able to derive depths from dark and bright melt ponds. However, in the results, it looks like the R2 for the dark ponds is much worse than for bright ponds. It is misleading to claim that their model is accurate for dark ponds. | As we pointed out in our first reply and in the reply to Referee #1, we agree that "universal" is not the correct term and replaced it with "applicability". As proposed in our previous reply we expanded the discussion according to your comment (Section 4.2). We point out that R2 is much worse for the dark pond but discuss that this is due to the small number of samples and the shallowness of the measurements. The similar RMSE in dark and bright ponds proves that the model's accuracy is similar in both cases, which is also illustrated in Figure 11B. |
| Likewise, the authors state that their study provides "the most comprehensive set of Rrs and depth measurements from Arctic melt ponds...". However, the study only presents 49 depths from three melt ponds with a depth range of 6 to 25 cm. A quick search of the literature reveals that this statement misleading. Malinka et al.(2018) https://doi.org/10.5194/tc-12-1921-2018 used coincident depth and spectra measurements from three different areas of the Arctic (SHEBA experiment in 1998, Barrow in2008 and the Polarstern in 2012). Tedesco and Steiner (2011) | Thanks to your valuable comment, we revisited the studies you mentioned and restructured the introduction and discussion sections, which improved the quality of the manuscript a lot. We integrated the work of Legleiter et al. (2014) and Tedesco and Steiner (2011) into the introduction emphasizing the difference between ponds on sea ice and glaciers and the consequences for model development. In the discussion section, we refer to the albedo data sets from melt ponds on sea ice and explain that a comparison of R and Rrs is not trivial due to the non-lambertian behaviour of the water |

| | |
|---|---|
| doi:10.5194/tc-5-445-2011 and Legleiter et al.(2014) doi:10.5194/tc-8-215-2014 collected hundreds if not thousands of coincident depth and spectra measurements in a melt pond on the Greenland Ice Sheet that had depths of up to 10 m. The authors should review this literature (including those studies from the Greenland Ice Sheet) before making such claims. | surface.
We deleted the paragraph you were referring to because it only caused confusion without really contributing to the manuscript. Instead, we placed the statement in the discussion section where we use it to provide reason for more measurements in order to overcome the limited variability in our test data. |
| The problems outlined above raise questions about the novelty and significance of the study. The only real result in the abstract is that the "…results indicate that pond depth is retrievable from optical data under clear sky conditions". As far as I understand (and based on the references in the previous paragraph) this is not a novel finding. The authors should think more deeply about how their study advances our understanding and build on previous research. However, in the present version, this study may only have limited interest to the cryospheric community | Thanks to your thorough review of our manuscript and the reference to other studies, we were able to restructure and improve our manuscript. In the updated abstract and conclusion we explicitly point out the novelty of our approach which we also describe at the beginning of this letter.
The three novelties are:
   (i)    The utilization of near-infrared light for pond depth retrieval with unprecedented accuracy.
   (ii)   The adjustment to changing sun zenith angle.
   (iii)  The semi-empirical nature of the model, i.e. training on simulated data and testing on field data.
We further hypothesize that the model may improve the bathymetry mapping in shallow water areas of supraglacial lakes. |
| The writing style is vague in many places. For example: L5: "Key elements" L33:"easy-to-use" L96: "expert knowledge" L234: "hitting the same spot" L235: "tricky" L245:"widely independent" The authors should consider being more specific where possible to improve the readability | We tried to be more specific and rewrote the examples you mentioned where possible. Likewise, we tried to be as specific as possible in the new paragraphs of the updated manuscript. |

**Specific comments**

We addressed most of the comments as proposed in our first reply.

| | |
|---|---|
| L28-209: Considering the apparent similarity between this work and Malinka et al. (2018), the authors should consider adding a much more thorough description of how the two models differ in the introduction. | In the updated introduction, we expanded the description of the model described in Malinka et al. (2018) and point out the major differences. |
| P126: Please justify why pond depth is extrapolated to 1 m when the maximum pond depths in this study were 25 cm. | The intention of our study was the development of a model that allows pond depth retrieval from remote sensing data. Therefore, it needs to be applicable to ponds in all states of the melting process. Ponds on sea ice are usually shallower than 100 cm because pond depth is restricted by ice thickness. |

| | It was not our aim to find the optimal model for our field data set. Instead, we wanted to develop general approach that is transferrable. We therefore generated a spectral library for model training and used the field data for testing. |
| --- | --- |

[revised manuscript text omitted]

---

## Author Response (AR2)

Dear Stef Lhermitte and anonymous referees,

Thank you very much for reviewing our manuscript. We are happy that the changes we have made to the first version of the manuscript have improved its quality and that it is now accepted for publication with subject to minor revisions.

Please find our answers to your latest comments and the corresponding changes in the following table.

Best regards,

Marcel König and Natascha Oppelt

**Editor**

**Minor comments**

| Comment | Answer |
|---|---|
| Be more upfront about accuracy and limitations. If your method only works for SZA's between 58.9-61deg, it is actually very limited for practical use. I think this limitation, but also the others highlighted by reviewer-2, are important to clearly communicate. | The method works for every possible sun zenith angle but has only been validated for SZAs between 58.9 and 61 deg. As we outlined in L215f the model is fit on seven spectral libraries and corresponding SZAs between 0 and 90 deg. We use Richards curves to retrieve the slope and offset of the linear model for every possible SZA. However, due to the limited amount of field data we could only validate the model for SZAs between 58.9 and 61 deg. We therefore point out the need for more data. To clear up the confusion we changed L319f to

"We therefore conclude that, the present model is only valid for clear sky conditions. The model accounts for the influence of varying solar zenith angles but field data was limited to solar zenith angles between 58.9° and 61°. To enlarge its validity range more field data covering different weather and illumination conditions is necessary". |
| Abstract and text: please try to be consistent in reporting r or R^2. On line 15 (abstract) but also elsewhere you sometimes report r and on other locations R^2 | We used Formula 10 as recommended by Kvålseth (1985) and implemented in scikit-learn. Please note that $R^2$ is not Pearson correlation coefficient squared. Instead $R^2$ is a measure of how much of the total variation of the measured data (about its mean) is explained by the model. $R^2 = 1$ means that the model is a perfect predictor, $R^2 = 0$ means that the sample mean is an equally good predictor, and $R^2 < 0$ means that the model as a predictor is worse than just using the sample mean. |
| L36: "enable monitoring of pond water characteristics" -> remove a | Thanks, we removed a. |
| L70: "the range of depths": depths instead of depth? | Yes, we changed it accordingly. |

**Referee #1**

**Minor comments**

| Comment | Answer |
|---|---|
| L 60: "acquired under and different illumination conditions" ?? | Thanks. Changed to "acquired under different illumination conditions". |
| L83: "with a spectral sampling rate < 1 nm and an spectral resolution of 3.0 nm"? It is not clear what this specification means. Is the light sampled with detectors spaced by 3.0 nm and then interpolated to 1 nm? | Spectral resolution defines the narrowest spectral feature that can be resolved. Sampling rate is the separation between two adjacent sampling bands. As this caused confusion, we remove the sampling rate information and change the sentence to "with a spectral resolution of 3.0 nm" (for the Ocean Optics, L83) and "with a spectral resolution of 1.8 nm" (for the Ibsen, L91). |
| L154: " We therefore defined a pure water column without absorbing or scattering water constituents and computed remote sensing reflectance in shallow water above the water surface…" so no absorption in the pond water? | Right. We account for the absorption of water but not for the absorption of water constituents (e.g. CDOM). We changed the sentence to "We therefore defined a pure water column without additional absorbing or scattering water constituents…". |
| L218: (and other instances): "solar" zenith angles, instead of "sun" zenith angles | We changed sun to solar throughout the manuscript. |
| L261: "Converting albedo to radiance is only possible when assuming a Lambertian scattering behaviour." This sentence needs to be reworded. If I understand correctly, what the authors are trying to say is "It is only appropriate to derive an accurate radiance directly from the albedo of a Lambertian surface." | Thank you for rewording the sentence, we changed it accordingly. |
| L294: "The data set, however,origins from only three ponds, …" originates | Thank you. |
| L322: "separation of model calibration on simulated data" not sure what this phrase means. It needs to be rewritten. | We changed the sentence to "The model calibration on simulated data and independent validation on in situ data proves the applicability and robustness of our approach". |
| L336: "We further assume that the additive signals of the water surface and the atmosphere on the spectrum measured at a remote sensor may complicate the retrieval of pond depth." This sentence needs to be rewritten. Its meaning is not clear—of course the atmosphere will modulate the signal, but its not clear why the signal of the water surface is a potential problem. | We changed the sentence to "We further assume that the additive signals of the atmosphere and reflections of skylight at the water surface may complicate the retrieval of pond depth with remote sensors". |

**Referee #2**

**Minor comments**

| | |
|---|---|
| I am happy to find that the authors took onboard many of my comments. The additional text in the introduction provides much better justification for the study. The revised manuscript also contains a much more thorough review of previous literature, including some relevant research from the Greenland Ice Sheet. My one remaining concern is that the authors overstate the accuracy of their method. I would like to see some additional caveats included in the abstract and conclusions. For example, while it may be true that the errors are 2-3 cm (the uncalibrated error is 3.29 cm and the calibrated error is 2.81 cm) the ponds are extremely shallow. Looking at Figure 11, it appears that pond depths average around 16 cm which would equate to a 20% error. I encourage the authors to be upfront about their accuracies in the abstract (L14) and conclusions (L352) by presenting the errors as percentages as well as absolute values. I would also encourage the authors to add the word "shallow" in the abstract since the depths of melt ponds sampled were small (somewhere in L13-15). Below are some more specific comments. | We added the formula for the normalized root mean square error (nRMSE) to the manuscript (now Formula 12), computed the nRMSE in % for the results and included the percentage values in the abstract and conclusion of the manuscript. We also added „shallow" to L14. |
| L28: Suggest replacing "been put on" with "investigated". | We changed the sentence according to your suggestions to "Recent efforts were made to observe the evolution of melt pond fraction with satellite data but few studies investigated melt pond depth …". |
| L33: "reckon" is informal, consider replacing. | We changed „reckon" to „point out". |
| L129: Typo, "the" instead of "he" | Thanks for spotting. |
| L303-311: The 3.05 cm error in dark ice melt ponds as shallow as 6 – 14 cm is still approximately 20-30%? | We added the nRMSE of 35 % to L293. |
| L307: How can the R2 be less than 0? | We used Formula 10 as recommended by Kvålseth (1985) and implemented in scikit-learn. Please note that R2 is not Pearson correlation coefficient squared. Instead R2 is a measure of how much of the total variation of the measured data (about its mean) is explained by the model. R2 = 1 means that the model is a perfect predictor, R2 = 0 means that the sample mean would be an equally good predictor and R2 < 0 means that the model as a predictor is worse than just using the sample mean. |

| L314-315: Could the authors elaborate a bit more about solar zenith angles? How many days per year do solar zenith angles reach 58.9 and 61? Is it just July? Or a longer period? | The model itself is fit to solar zenith angles between 0 and 90 degrees. Yet, due to the limited amount of data, we were only able to validate it for solar zenith angles 58.9 and 61 degrees. We therefore point out the need for more data.
 To clear up the confusion we changed L319f to

 "We therefore conclude that, the present model is only valid for clear sky conditions. The model accounts for the influence of varying solar zenith angles but field data was limited to solar zenith angles between 58.9° and 61°. To enlarge its validity range more field data covering different weather and illumination conditions is necessary". |
| --- | --- |

**Other changes**

We found some typos ourselves and added a paragraph to the acknowledgement in order to acknowledge the financial support of Land Schleswig-Holstein.

[revised manuscript text omitted]